# Silencing the Rainbow: Prevalence of LGBTQ+ Students Who Do Not Report Sexual Violence

**DOI:** 10.3390/ijerph20032020

**Published:** 2023-01-22

**Authors:** Heather Tillewein, Namrata Shokeen, Presley Powers, Amaury J. Rijo Sánchez, Sasha Sandles-Palmer, Kristen Desjarlais

**Affiliations:** 1Department of Health and Human Performance, Austin Peay State University, Clarksville, TN 37044, USA; 2Department of Sociology, Monk Prayogshala, Mumbai 400072, India; 3Department of Sociology and Anthropology, Middle Tennessee State University, Murfreesboro, TN 37132, USA; 4Department of Sociology, University of Illinois at Urbana-Champaign, Urbana, IL 61801, USA; 5California School of Professional Psychology, Alliant International University, San Francisco, CA 91803, USA; 6Department of L’Nu, Political, and Social Studies, Cape Breton University, Sydney, NS B1M 1A2, Canada

**Keywords:** sexual violence, LGBTQ+, sexual orientation, higher education, non-reporting

## Abstract

Previous research on sexual violence suggests that there is a higher likelihood of students from LGBTQ+ community experiencing sexual violence and not reporting it. This study investigates various types of sexual violence experienced by the LGBTQ+ students and the prevalence of reporting it. The study further determines why different types of sexual violence are not being reported. This study uses a LGBTQ+ scholarship survey data conducted in 2019. Using Pearson’s chi square analysis, the study investigates the relationship between who experienced various kinds of sexual violence and those who do not report it. The study provides descriptive analysis of reasons for not reporting sexual violence across different types of sexual violence. The results show that there is a statistically significant relationship between those who experienced various kinds of sexual violence and those who do not report it. In addition, the study illustrates mistrust in support services and fear of embarrassment as the major reasons resulting in non-reporting behaviors. In conclusion, the study illustrates high prevalence for various types of sexual violence against LGBTQ+ students as well as high underreporting. Study results have implications for health professionals and institutions to focus efforts in making school environments safe and inclusive for LGBTQ+ students.

## 1. Introduction

LGBTQ+ post-secondary student reports of sexual violence have received less attention than those of cisgender and heterosexual students in the U.S. While sexual violence reporting in US postsecondary schools have been steadily on the rise throughout the 2000s, and the federal government recognizes sexual violence as a public health issue, little is known about LGBTQ+ student experiences of sexual violence. The scant data on these experiences suggest high rates of sexual violence for genderqueer and sexual minority students. Multiple data sources, including The Association of American Universities (AAU) Campus Climate Survey have demonstrated risk factors associated with sexual violence, including being a woman, self-identifying as TGQN (transgender, genderqueer, gender questioning, nonbinary), and enrolling as an undergraduate student (when compared to either graduate or professional students) [1]. 

Understanding the risk factors associated with sexual violence within educational institutions is crucial to successful intervention, including prevention. The National Institute of Justice (NIJ) reports alcohol use as a risk factor for sexual violence based on findings from the 2007 Campus Sexual Assault Study [2]. At least half of reported assaults among college students involved alcohol consumption by either the perpetrator, the victim, or both [2]. Other factors associated with risk of victimization include sorority membership, number of sexual partners, freshmen or sophomore status, and social setting [2]. Unlike the AAU’s questionnaire, The NIJ’s identification of risk factors focused primarily on women without specification of gender identity or sexual orientation. While the study highlights the importance of some key risk factors, its lack of specificity about gender identity and sexuality may obscure the sexual violence experienced by genderqueer and sexual minority women.

The literature on sexual violence risk factors for LGBTQ+ college students is scant, but some has identified risk factors of sexual violence for the broader LGBTQ+ population. For example, The Human Rights Campaign Foundation reports poverty, stigma, and marginalization increase sexual violence risk within the LGBTQ+ community [3]. Hate-motivated crimes, which LGBTQ+ individuals are more likely to face than non-LGBTQ individuals, can also take the form of sexual violence [3]. Furthermore, Sutton and colleagues suggest sexual violence victimization for LGBTQ+ students is bolstered by high school experiences of sexual violence and excessive alcohol use [4]. The emotional and academic impact of sexual violence on LGBTQ+ students significantly place these students at greater risk for mental health issues, lower school attendance and overall poorer academic performance [5,6]. 

### 1.1. Experiences of Sexual Violence against the LGBTQ+ Community within Post-Secondary Education

Research on LGBTQ+ youth has highlighted that LGBTQ+ students are at greater risk of experiencing several forms of violence than are their cisgender and/or heterosexual counterparts. Extensive research conducted through the Gay, Lesbian, and Straight Education Network (GLSEN) documents the experience of violence for LGBTQ+ ranging from different forms of exclusion, discrimination, harassment, and assault at increasingly alarming rates [7]. Sexual harassment and assault comprise great part of the types of violence LGBTQ+ students experience [7,8]. In 2019, the GLSEN report identified that 86.3% of LGBTQ students experienced harassment or assault based on their identity or sexual orientation, and 58.3% were sexually harassed. Further, this report has continuously shown that the experience of violence, ranging from verbal harassment and bullying to psychological and physical abuse presents considerable barriers for LGBTQ+ students to fully engage with their school environment and educational purposes [7]. The results of these experiences have both short- and long-term impact contributing to societal problems such as drug abuse and psychological problems. 

Research on violence against queer people has explored how anti-LGBTQ+ violence is experienced differently according to other intersecting social locations. Thus, sexual violence occurs on an axis of both gender and sexuality, and expands beyond other markers such as race, class, age, immigration status, and/or ability. The experience of sexual violence can be understood as a gendered issue threatening feminine and female bodies at higher rates. Further, racialized bodies are also at higher risk of experiencing multiple forms of anti-LGBTQ+ violence, including sexual violence [9]. A study comprising forty-seven in-depth interviews with people who encountered violence for being perceived as a member of the LGBTQ community showed, lesbians experienced several forms of sexual harassment and assault; moreover, a type of violence molded by their status as women in a society dominated by men [9]. 

The experiences of anti-LBGTQ+ sexual violence become two-folded when considering racial and ethnic minorities positing transwomen of color as the most vulnerable within these populations [7,10,11]. Experts state that LGBTQ+ youth of color confront a ‘tricultural’ experience as they face homophobia from their respective racial or ethnic group, racism from within a predominantly white LGBT community, and a combination of the two from society at large [10]. When individuals are exposed to compounded forms of discrimination, the dehumanizing effects of this can make individuals more vulnerable to the short- and long-term impacts of sexual violence than their non-racialized and cisgender counterparts. A study conducted to assess Black transgender women in community institutions found that, transphobic and homonegative responses of school administrators and officials had violent and deterrent effects on black transgender students [12]. These effects not only harmed the mental and emotional health of these students but normalized throughout the student body the discriminatory treatment of these students, fostering a hostile environment for LGBTQ+ students all around. These experiences pose multiple threats to LGBTQ+ youth in schools and further discourage students to question their experiences and diverge from cis- and heteronormative expectations. 

In a longitudinal study measuring the correlation between school belonging, depressive symptoms, and sexual harassment victimization among LGBTQ+ high school students in U.S. schools reiterates how the experiences of sexual harassment have detrimental effects on LGBTQ+ high schoolers’ mental health and overall school outcomes [8]. Further, this study finds sexual victimization towards LGBTQ+ students have effects of long-term suffering and diminished ability of developing a healthy sense of school belonging. Despite the striking evidence made available about the higher likelihood of experiencing sexual violence by the LGBTQ+ students’ community, much of the literature on student sexual assault victims focuses on the phenomenon from a sex and gender normative perspective, leaving much to be examined when understanding sexual violence among LGBTQ+ victims [13].

### 1.2. LGBTQ+ Sexual Violence and Under-Reporting

In between the ignored story of sexual violence against LGBTQ+ students and the elevated risk of being sexually abused among them, one of the major quandaries among LGBTQ+ sexual violence victims remains whether to disclose their experience or not [14]. Current data show that a larger portion, as high as 90%, of student sexual assault victims, do not report it, which may support the literature on the high prevalence of sexual assault on college campuses [15]. The Rape, Abuse & Incest National Network, RAINN, report suggests that 26 percent of females between 18–24 years did not report sexual violence because they believe it is a personal matter [16]. Nearly 9% did not report as they believed that the support services could not do anything to help. With respect to LGBTQ+ students’ community, various reports suggest that LGBTQ+ students are even less likely to report sexual violence than their heterosexual peers [17]. 

Todahl and colleagues assessed the existing responses of LGBTQ+ students to sexual violence and identified various reasons that may contribute to low reporting (i.e., Fear of being judged due to sexuality, victim blaming, lack of available resources or lack of trust in authorities) [18]. Homophobia, stigma and discrimination against LGBTQ+ students are quite prevalent in the U.S education system [19]. Such negative attitudes toward LGBTQ+ students often lead to feelings of being rejected by friends, family and peers, and fear of being bullied, teased, and harassed if they disclose their identity [20]. In this line, Weiss reports that sexual crime victims are more susceptible to feelings of shame compared to victims of other crimes [20]. Similarly, findings from Felson and Pare also suggest underreporting of sexual assault has more to do with victims feeling embarrassed, fearing retaliation, and believing police involvement will not help them than inhibitions related to the victim’s gender [13]. Individuals who have been sexually assaulted also experience enhanced internal feelings of shame of being a sexual minority and feelings of shame and rejection. Consequently, LGBTQ+ survivors may fear their sexual orientation or gender identity may become the focus of attention or even the perceived cause of the violence rather than the violence itself [21]. 

LGBTQ+ survivors also report being further harassed as a result of the sexual violence and are exposed to associated risks related to their housing, employment, and faith community, as well as negative response from family and acquaintances [22,23]. When asked why they did not seek help, sexual minority respondents were nearly three times more likely than heterosexual respondents to report that they thought they would be blamed for the sexual violence [24]. As a consequence, such negative feelings and outcomes might refrain LGBTQ+ students from reporting sexual violence. In addition, the increasing cases of hate crimes based on sexual orientation and the victim-blaming culture against sexual violence victims further make it difficult to report [25]. 

Literature on non-reporting of sexual violence against LGBTQ+ students further show the fear of the perpetrator. The power differences and heteronormative norms of masculinity can also influence one’s decision to report sexual violence. GSLEN (2019) shows how almost 70 percent of LGBTQ+ students stated that one of the prominent reasons for non-reporting is being threatened by the perpetrator. Reporting becomes even more difficult if the sexual violence perpetrator is known to the victim [13].

This fear of the perpetrator or being judged also stems from the lack of trust in the support services and community systems. The GLSEN (2019) report shows how most of the U.S policies and laws are unable to protect the LGBTQ+ student populations [7]. Research suggests that a key problem faced by LGBTQ+ sexual violence survivors is that many existing programs and services have been designed from a cisnormative and heteronormative perspective. This in turn may generate mistrust and decreased help-seeking by LGBTQ+ sexual violence victims. In a study conducted by Todahl and colleagues, through qualitative interviews they show the existence of low community awareness and support for sexual violence in a LGBTQ+ community [18].

Sexual violence against the LGBTQ+ students’ community is often surrounded by negative societal responses and LGBTQ+ specific rape myths [26]. While a few existing studies observe the higher victimizations of sexual violence as well as non-reporting of among LGBTQ+ students, the notions and images of sexual assault are often guided by heterosexual norms of what constitute sexual violence. It is important to stipulate the variety of sexual violence acts, including sexual slurs, assaults, forced online sexting and to better understand the prevalence perpetrated against the LGBTQ+ students’ community alongside the myriad of reasons for not reporting such instances. This will account for the various types of sexual violence and uncover the extreme range of sexual violence as well as the reason for non-reporting that often remain unexplored in academia. Furthermore, this will allow disclosing the even higher prevalence of sexual violence among LGBTQ+ students. 

### 1.3. The Purpose of the Study

The purpose of this study was to investigate the types of sexual violence experienced by LGBTQ+ post-secondary students, prevalence of students. Reporting and reasons for not reporting. This study provides insights as to the phenomenon of sexual violence not being reported and hopes to provide recommendations for public health professionals to create interventions to target this issue. 

## 2. Materials and Methods

### 2.1. Study Design

The study was a quantitative study, using a descriptive cohort design, that investigated incidence of various types of sexual violence experienced by LGBTQ+ individuals and the prevalence of those not reporting the sexual violence. This instrument measured various types of sexual violence and those who did not report at community college/vocational school, four-year college, and graduate school. This study looked at various sexual violence and the prevalence of reporting during the 2019 academic year. The Access to Higher Education Survey instrument used for data collection was developed through a community advisory board made up of 30 interviews from expert professors and policy advocates. Researchers developed a survey for a LGBTQ+ scholarship foundation in regard to the population’s experiences in higher education. Data were collected through nine data collection cycles and the survey was administered in 2012–2021. This study specifically looks at the 2019 survey before COVID-19, and all the data were collected using SurveyMonkey, an online survey platform. 

### 2.2. Participant Recruitment

Participants were recruited from a pool of LGBTQ+ higher education scholarship applicants. Potential participants received an email from Point Foundation with a link to the study. Emails were sent in waves, with wave 1 the recruitment started in the beginning of the year. The second wave for recruitment happened in February. Those individuals who made it to the semi-finalist stage of the Point Foundation scholarship would receive another email for recruitment. The survey was anonymous and to maintain anonymity, the survey did not ask for identifiable information. Additionally, no institutional information was collected to maintain confidentiality. Undocumented students and international students were eligible to participate in the survey. There were separate scholarships for community college/vocational college students, undergraduate students, and graduate students. 

### 2.3. Participant Demographic

From the 2019 data set (*n* = 808), there were 26.36% (*n* = 213) individuals that identify as gay, 24.13% (*n* = 195) identify as bisexual, 23.89% (*n* = 193) identify as queer, 13.99% (*n* = 113) identify as lesbian, 9.65% (*n* = 78) identify as other, and 1.98% (*n* = 16) identified as heterosexual/straight (Table 1). Out of the total participants, 58.91% did not use the any of the gender identity terms, 9.16% identified as transgender man, 7.55% identified as other, 6.31% identified as genderqueer, 5.69% identified as gender nonconforming, 3.96% identified as fluid, 2.85% identified as transgender, 1.61% identified as androgynous, and 1.36% identified as female to male (Table 2). There was only 0.99% that identified as transgender woman, 0.87% identified as two-spirit, and 0.62% identified as male to female. 

Questions pertaining to demographics of the participants were placed at the end of the survey, for the age of the participant question only 789 responses were reported due to survey fatigue. The age of participants (*n* = 789) was 34.85% (*n* = 282) were 18–20 years of age, 32.70% (*n* = 264) were 21–30 years of age, 23.32% (*n* = 188) were less than 18 years of age, and 6.46% (*n* = 52) of participants identified as 31–40 years of age (Table 3). There were 19 individuals who did not respond to the question of age. The race of participants was 50.9% identified as Caucasian, 16.63% identified as Latino, 10.46% identified as Black or African American, 7.37% identified as Biracial/Multiracial, 7.27% identified as Asian, Pacific Islander, or Indian, 3.98% identified as Native American (including Alaska or Hawaii), 1.89% identified as other, and 1.49% preferred not to identify as race (Table 4). 

### 2.4. Data Analyses

The Access to Higher Education Survey includes 179 survey items measuring various aspects of the LGBTQ+ experience. This study focused on the relationship between those who experienced different categories of sexual violence and whether or not they reported the incident. The specific questions asked were: “Since you have been a student, has a student or someone employed by or otherwise associated at your college made sexual remarks or told jokes or stories that were insulting or offensive to you?”. Similarly, the survey gives options for multiple types of sexual violence (Table 5) experienced or never experienced by the participants. The survey also asked “Did you report any of these occurrences to campus officials or police?”. The question gives options whether the survey participants have reported or not reported the following type of sexual violence or not, i.e., made sexual remarks or told jokes or stories that were insulting or offensive to you?; made inappropriate or offensive comments about your or someone else’s body, appearance or sexual activities?; said crude or gross sexual things to you or tried to get you to talk about sexual matters you didn’t want to?; emailed, texted, tweeted, phoned, or instant messaged offensive sexual remarks, jokes, stories, pictures or videos to you that you did not want?; continued to ask you to go out, get dinner, have drinks or have sex even though you said, “No”?; sexually assaulted you (e.g., unwanted sexual touching, forced or pressured into having sex). 

Furthermore, to determine the reasons for not reporting the specific type of sexual violence the survey asked “Why did not you report?”, with an option to select all the choices that applied. The survey gives the following options, i.e., Embarrassed or ashamed; Did not think any action would be taken; Thought you would get in trouble; Did not want to get the other person in trouble. The same options were given for each type of sexual violence, i.e., made sexual remarks or told jokes or stories that were insulting or offensive to you?; made inappropriate or offensive comments about your or someone else’s body, appearance or sexual activities?; said crude or gross sexual things to you or tried to get you to talk about sexual matters you didn’t want to?; emailed, texted, tweeted, phoned, or instant messaged offensive sexual remarks, jokes, stories, pictures or videos to you that you didn’t want?; continued to ask you to go out, get dinner, have drinks or have sex even though you said, “No”?; sexually assaulted you (e.g., unwanted sexual touching, forced or pressured into having sex)? 

The paper uses STATA version 14.1 to measure the Pearson’s chi-squared test that is used to determine whether there is a statistically significant difference between those who experienced different categories of sexual violence and those who did not report the violence. Furthermore, frequency tables were tabulated for each type of sexual violence and the reasons for not reporting them.

## 3. Results

A Chi-Square was performed to determine the relationship between those who experienced different categories of sexual violence and the reasons that they did not report the violence (Table 6). There was a significant relationship between “those who said crude or gross sexual things to you or tried to get you to talk about sexual matters you didn’t want to?” and those who did not report this type of violence, X^2^ (1, *n* = 181), 8.382, *p* = 0.004. Meaning that those who experienced this type of sexual violence were more likely to not report. Out of 77 participants who experienced this type of sexual violence, 92.25% (*n* = 71) of those who had experiences this type of sexual violence did not report. There was also a significant relationship for the variable “emailed, texted, tweeted, phoned, or instant messaged offensive sexual remarks, jokes, stories, pictures or videos to you that you didn’t want?” and those who did not report, X^2^ (11, *n* = 178), 10.443, *p* = 0.001. This relationship means that those who experienced this type of sexual violence were more likely to not report the incident. Out of 39 participants who experienced this type of sexual violence, there were 87.2% (*n* = 34) of individuals who had experienced the sexual violence and did not report. Additionally, there was a significant relationship between the variable “made sexual remarks or told jokes or stories that were insulting or offensive to you?” and those who did not report the incident, X^2^ (1, *n* = 184) 5.3125, *p* = 0.021. This means that those who experienced this type of sexual violence were more likely to not report. Out of 106 participants who experienced this type of sexual violence, 90.5% (*n* = 96) individuals who had experienced this form of sexual violence and did not report. 

A Chi-Square Independence Test was also performed to determine the relationship between the variable “continued to ask you to go out, get dinner, have drinks or have sex event though you said “No”?” and those who did not report, X^2^ (1, *n* = 177), 7.868, *p* = 0.005. This means that those who experienced this type of sexual violence were more likely to not report the incident. Out of 35 participants who experience this type of sexual violence, 91.4% (*n* = 32) who had experienced this type of unwanted sexual violence and did not report the sexual violence. Additionally, a Chi-Square was used to determine the relationship between “sexually assaulted you (e.g., unwanted sexual touching, forced or pressured into having sex)?” and those who did not report the sexual assault, X^2^ (1, *n* = 177), 13.320, *p* < 0.001. This relationship means that those who experienced being sexually assaulted were less likely to report the assault. Out of 32 participants who experienced this type of sexual violence, 87.5% (*n* = 28) of participants who experienced this type of sexual violence and did not report the incident. 

A Chi-Square was used to determine the relationship between the variable “made inappropriate or offensive comments about you or someone else’s body, appearance, or sexual activities” and those who did not report, X^2^ (1, *n* = 183), 2.196, *p* = 0.138. The analysis determined there was no significant relationship between the two variables. Out of 111 participants who experienced this type of sexual violence, it was determined that 91.9% of participants that experienced this did not report, but the analysis determined it was not significant. 

Table 7 shows the frequency distribution of various types of sexual violence and reasons for not reporting a particular type of violence. The data show that among varied types of sexual violence the majority of respondents showed their mistrust in support services. Among those who experienced sexual remarks or were told jokes or stories that were insulting or offensive to them, 75% (*n* = 72) did not report because they did not think any action would be taken. Similarly, among those who experienced any kind of offensive comments about their or someone else’s body, appearance or sexual activities, nearly 69% (*n* = 71) did not think any action would be taken. Nearly 67% and 73.5%, respectively, of those who experienced crude or gross sexual things and emails, texts, tweets, phone or instant messages, offensive sexual remarks, jokes, stories, pictures or videos that the respondent did not want. Furthermore, more than half (*n* = 16) of those who experienced any kind of sexual violence did not think any action would be taken if they reported the sexual violence. A fairly significant number of participants also felt that if they reported any type of sexual violence, they will face embarrassment. Feeling embarrassed as a factor of non-reporting behaviors was highest (57.14%, *n* = 16) among those who experienced sexual violence, followed by 25.35% those who experienced crude or gross sexual things that they did not want and nearly 20% each who experienced offensive comments on their body and continued to ask you to go out, get dinner, have drinks or have sex event though the participant said “No”, respectively. In addition, although small in numbers, but amongst all types of sexual violence victims many thoughts reporting a particular type of sexual violence might get them into trouble. The common and highest reason among all categories of sexual violence for not reporting was that participants just did not think any action would be taken. 

## 4. Discussion 

In line with previous research, the present study also shows that there is a high prevalence of LGBTQIA+ students who have been a target of sexual violence and that with this high incidence of sexual violence. There is also a high prevalence of individuals not reporting the sexual violence regardless of the severity of the action. Only those participants who had experienced inappropriate comments or offensive comments about themselves or someone else’s body, appearance, or sexual activities had no significance on not reporting the incident. However, the data showed that the majority of those who had this experience did not report the incident. The high prevalence of non-reporting of sexual violence can be attributed to several socio-cultural norms and administrative and systemic reasons. For example, the present study shows that LGBTQ+ students who are victims of sexual violence do not report because they do not want to be blamed or feel embarrassed, many think that any action would be taken, some do not want the other person to be in trouble and some also think that reporting might even get them into trouble. The results from the study show that majority of individuals do not report as they do not think there would be no action taken against the perpetrator. The present study shows much higher level of mistrust in support services compared to heterosexual female students (only 9%) mentioned in the RAINN report [16]. The findings from the study further questions the workings and functioning of support services both inside and outside of the campus. The study highlights the need for more inclusive support services to address sexual violence against LGBTQ+ students at the campus. There is also a need to build and strengthen communities that support that assist and educate the survivors of sexual violence about the formal processes of filling complaints. Survivors of these sexual violence reported being embarrassed as another reason for not reporting sexual violence. This phenomenon could be due to the homonegativity or social norms of masculinity that individuals are ashamed to report sexual violence which these negative feelings that lead to LGBTQ+ students fear of being rejected, bullied, teased, or harassed [7]. The study showed that fear of being embarrassed and no action being taken against the aggressor has been why survivors do not report. This has also been discussed in previous studies that victims underreport due to feeling embarrassed, fearing retaliation, and not being believed due to sexual and gender identity [13]. This is mainly because of a culture of “blaming the victim”, where those who experience sexual violence, themselves are considered responsible for the violence inflicted. Victim-blaming is often well-rooted in heterosexual and patriarchal socio-cultural norms. Hence, in terms of social and cultural implications, this study highlights a need for awareness of the unique and intersectional experiences of LGBTQ+ students; specifically, as these experiences pertain to ingrained socio-cultural norms that guide administrative policy at most educational institutions. Without proper representation, viewpoints of LGBTQ+ students and individuals are excluded, perpetuating a culture of heteronormative values. As such, acts of sexual violence will continue to go unreported and the experiences of LGBTQ+ students will not be captured. By intentionally combating systemic inequality in educational institutions, an inclusive culture can be cultivated and LGBTQ+ students can receive the support they need to feel comfortable as well as heard when reporting instances of sexual violence. 

## 5. Limitations

A limitation to this study is there was a lack of diversity of minority sexual identities and gender identities. Another limitation is the sensitive nature of the questions asked to assess sexual violence. These questions could be difficult or uncomfortable for participants to answer, thus making it difficult to know the true prevalence of sexual violence among LGBTQ+ individuals in higher education. Another challenge was having participants that were recruited not completing the survey or not starting the survey. Some types of violence, such as sexual remarks or comments, may not be reported to police. 

## 6. Conclusions

Research has shown that LGBTQ+ students face higher prevalence of sexual violence and underreports sexual violence incidents. Regardless of type of sexual violence, many LGBTQ+ individuals do not report due to embarrassment or believing that no action would be taken against the perpetrators. Survivors of sexual violence face negative health outcomes such as depression, anxiety, and negative mental health. Health professionals and academic institutions should focus efforts in making school environments safe and inclusive for LGBTQ+ students. Collaborative efforts can help reduce the prevalence of sexual violence on campuses and help students feel comfortable reporting sexual violence. Further, these efforts should be reflected in legal policies that both protect the livelihood of LGBTQ+ students and adequately sanctions perpetrators for their destructive and exclusionary behavior. 

## Figures and Tables

**Table 1 ijerph-20-02020-t001:** Frequency of gender identity.

Gender Identity	*n* (%)
I do not use these terms	476 (58.91%)
Female to Male (FTM)	11 (1.36%)
Male to Female (MTF)	5 (0.62%)
Transgender	23 (2.85%)
Transgender Man	74 (9.16%)
Transgender Woman	8 (0.99%)
Gender Nonconforming	46 (5.69%)
Genderqueer	51 (6.31%)
Two Spirit	7 (0.87%)
Third Gender	1 (0.12%)
Fluid	32 (3.96%)
Androgynous	13 (1.61%)
Other (please specify)	61 (7.55%)
Total	808 (100%)

**Table 2 ijerph-20-02020-t002:** Frequency of Sexual Orientation.

Sexual Orientation	*n* (%)
Heterosexual/Straight	16 (1.98%)
Gay	213 (26.36%)
Lesbian	113 (13.99%)
Bisexual	195 (24.13%)
Queer	193 (23.89%)
Other	78 (9.65%)
Total	808 (100%)

**Table 3 ijerph-20-02020-t003:** Frequency of Age.

Age	*n* (%)
less than 18 years	184 (23.32%)
18–20	275 (34.85%)
21–30	258 (32.70%)
31–40	51 (6.46%)
41–50	14 (1.77%)
51 and older	7 (0.89%)
Total	789 (100%)

**Table 4 ijerph-20-02020-t004:** Frequency of Racial Background.

Racial Background	*n* (%)
Native American (incl Alaska and Hawaii)	40 (3.98%)
Asian, Pacific Islander or Indian	73 (7.27%)
Biracial/Multiracial	74 (7.37%)
Black or African American	105 (10.46%)
Chicano, Hispanic, Latino	167 (16.63%)
White or Caucasian	511 (50.9%)
prefer not to respond	15 (1.49%)
Others	19 (1.89%)

**Table 5 ijerph-20-02020-t005:** Questions and Options analyzed from the survey.

Questions Analyzed for Determining Experiences and Reporting Different Type of Sexual Violence	Options Analyzed for Types of Sexual Violence
“Since you have been a student, has a student or someone employed by or otherwise associated at your college: Yes, or Never Experienced” “Did you report any of these occurrences to campus officials or police? Yes or No”	made sexual remarks or told jokes or stories that were insulting or offensive to you? made inappropriate or offensive comments about your or someone else’s body, appearance or sexual activities? said crude or gross sexual things to you or tried to get you to talk about sexual matters you didn’t want to? emailed, texted, tweeted, phoned, or instant messaged offensive sexual remarks, jokes, stories, pictures or videos to you that you didn’t want? continued to ask you to go out, get dinner, have drinks or have sex even though you said, “No”? sexually assaulted you (e.g., unwanted sexual touching, forced or pressured into having sex)?
Questions analyzed for determining reasons for not reporting (for each specific type of violence)	Options analyzed for reason for not reporting (for each specific type of violence)
“Why didn’t you report?”	Embarrassed or ashamed Did not think any action would be taken Thought you would get in trouble Did not want to get the other person in trouble.

**Table 6 ijerph-20-02020-t006:** Different Types of Sexual Assaults and Rates of Reporting.

	Report Status		
Types of Sexual Assault	Reported % (*n*)	Did Not Report % (*n*)	Total of Those Who Experienced the Type of Sexual Assault *n* out of 808
“made sexual remarks or told jokes or stories that were insulting or offensive to you?”	9.4% (10)	90.5% (96)	106
“made inappropriate or offensive comments about you or someone else’s body, appearance, or sexual activities”	8.1% (9)	91.9% (102)	111
“those who said crude or gross sexual things to you or tried to get you to talk about sexual matters you did not want to?”	7.8% (6)	92.25 (71)	77
“emailed, texted, tweeted, phoned, or instant messaged offensive sexual remarks, jokes, stories, pictures or videos to you that you did not want?”	12% (5)	87.2% (34)	39
“continued to ask you to go out, get dinner, have drinks or have sex event though you said “No”?”	8.6% (3)	91.4% (32)	35
“sexually assaulted you (e.g., unwanted sexual touching, forced or pressured into having sex)?”	12.5% (4)	87.5% (28)	32

**Table 7 ijerph-20-02020-t007:** Types of sexual violence that was not reported and reasons for not reporting.

	*n*	% of Cases
Types of Sexual Violence (Not Reported)		
Made sexual remarks or jokes	96	100%
Embarrassed	17	17.7%
did not think any action would be taken	72	75.0%
thought i would get into trouble	5	5.20%
did not want the person in trouble	11	11.45%
offensive comments on body	102	100%
Embarrassed	21	20.58%
did not think any action would be taken	71	69.60%
thought i would get into trouble	5	4.90%
did not want the person in trouble	10	9.80%
Said crude or gross sexual things	71	100%
Embarrassed	18	25.35%
did not think any action would be taken	48	67.60%
thought i would get into trouble	4	5.63%
did not want the person in trouble	8	11.26%
emailed or texted sexual things	34	100%
Embarrassed	5	14.70%
did not think any action would be taken	25	73.52%
thought i would get into trouble	4	11.76%
did not want the person in trouble	3	8.82%
continued to talk sexual things	32	100%
Embarrassed	7	21.87%
did not think any action would be taken	20	62.50%
thought i would get into trouble	3	9.37%
did not want the person in trouble	6	18.75%
Sexually assaulted	28	100%
Embarrassed	16	57.14%
did not think any action would be taken	16	57.14%
thought i would get into trouble	3	10.71%
did not want the person in trouble	5	17.85%

Note: The values are reported from multiple response questions, hence overall percentages may exceed 100 percent.

## Data Availability

The data presented in this study are available on request from the corresponding author. The data are not publicly available due to privacy and ethical considerations for the participants.

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
