# Peer review of "Silencing the Rainbow: Prevalence of LGBTQ+ Students Who Do Not Report Sexual Violence"

_ijerph, 2023, doi:10.3390/ijerph20032020_

Round 1

Reviewer 1 Report

Topic is relevant and important

The main critique I have is that its not clear to the reader what the study is about.  The intro is long and kind of rambling so it makes the reader confused as to the specific or main point of the paper.  It does read like this study was a dissertation.  But to make it a publication, it needs to get more focused.  Take out some content that is appropriate for a dissertation but is confusing in a manuscript.  There is a persuasive tone to this paper, to convince the reader that LGBTQ+ people are left out of research on sexual violence.  But then data is presented from research with LGBTQ+ people, which is counter to your point, they are not entirely left out. Pointing out gaps is important in a dissertation.  But in this paper, the purpose is not to establish that LGBTQ+ people are left out of the literature.  Stating once, that there are gaps in our knowledge regarding reporting and that this study is intended to fill the gap would be sufficient. 

Line 146-148, this is a provocative sentence that is not supported by the previous introduction.  Its not necessary either to set up the paragraph.

If the authors could tighten up the intro and delete some of it, staying on track with the topic – that LGBTQ+ people under report their experiences of sexual violence – would help the reader.

There are some typos throughout so a careful proofread should happen.

The first line of the paper reads as if the students were doing sexual violence to the academic institutions. It also is not supported by the content of the rest of the paragraph. There are statements about risk and prevalence of sexual assault for LGBTQ+ people which would appear to show that there is attention to this population in the literature.   This confusing first sentence sets the tone for the introduction – I don’t have a clear idea what the paper is about.  I think you should delete the sentence.

Line 93 – be careful about changing terms. You have used post-secondary academic settings, educational institutions, then this heading uses Education.  Later you use school violence – this term refers to a different location and also a different form of violence (sexual violence vs school violence)

Lines 93- 144 – this section should come out of the intro.  Some of the content could be moved to the discussion, if it is applicable to your findings.   This content provides background as to why people might not report.

The reason I recommend taking out this section, is because it is confusing the reader.  It takes the reader down a different road from the focus on college students and sexual assault.  This section is about high school and bullying.  It just confuses the reader as to what this study is about.  You are using a new term in this section too “school violence” and like I said, that takes the reader down a new road and is left unclear what the paper is about.

Line 66 – avoid words like “alarming.”  Conventionally, scientific writing avoids adjectives and adverbs like alarming.  Words like this are editorial and introduce some bias.  

Like 76-78 – this sentence can not be supported by the data and it also isn’t clear to the reader what you mean.  Are you saying that because so much attention has been on visibility of LGBTQ+ people that lack of data on gender or sexual orientation should be interpreted as everyone was cisgender?  Because if people were not cisgender then for sure their gender or sexuality would have been included?  That is a statement that you can’t support and so I think it should come out.

Line 198 – 209 should be deleted.  It just adds to lack of clarity about what the study is about.

The previous introduction included data about the prevalence and risk factors for sexual assault for LGBTQ+ people and included data specific to different populations under the LGBTQ+ umbrella.  To me, that shows that there is awareness of this issue.  Yes, gaps still exist.

Line 209 – Clearly state the purpose of the study and have the same clear statement in the abstract.

“The purpose of this study was to investigate the types of sexual violence experienced by LGBTQ+ post-secondary students, prevalence of students’ reporting and reasons for not reporting.”

Line 217 – This should be a statement of design, such as this study utilized a descriptive cohort design. 

In methods, labels interchange between sexual violence and sexual assault.  Sexual violence might be a better term – offensive comments are hurtful but not really an assault.  

The discussion is a place where you use both labels, sexual violence and sexual assault.  Like I said, hurtful words or insults that are about a person’s sexuality aren’t sexual assault.  They are incidences of abuse, hate motivated but I don’t know that they are sexual violence either.

A definition of sexual violence would help the reader.  And stick to a consistent term.

One limitation is the question, did they participant report to campus officials or police.  Hurtful comments probably wouldn’t get reported to police.  Students might report to a faculty member or advisor, but would someone answering that survey consider a teacher to be a campus official? I think there is some limitations with the survey itself  in terms of the wording of the questions.  And also in calling all of the violent experiences sexual violence.  Also, did they have the option for “other” as a choice in their list of reasons to not report? Did they give reasons for why they reported? 

Also consider the types of violence differently – the categories include instances of harassment/ bullying and partner violence. 

Results –

This section needs to get tighter too. 

In table 2, the Total column should show the number and percent of students who had this experience out of the total 808 students.  Presenting the data in that way shows that about around 10% of students had the first 3 things happen to them and lower percent had the last 3 things.

Are you able to report how many individuals had one of the types of violence? Some people might have experienced more than one type.   Of the 8?8 how many people had experienced at least one type of violence?

Discussion –

In your study only 4% reported sexual assault, this is lower than the 13% you have in the first sentence of the discussion.  Restatement of results is ok, but so what?  What are some measures that readers could or should take to use your results in their own work?  What are the implications of the findings?

You have an excellent topic and important data.  What does it mean for people at post-secondary institutions? What is being done to make things better?  What could be done to make things better?

Author Response

Thank you for the thoughtful feedback from the reviewers. Based on the reviewers’ recommendations, edits have been made to the submitted document as outlined below. The researchers again would like to thank the editorial board for allowing the researchers to make revisions and resubmit. Below I have outlined the specifies on how the comments of the reviewers were changed in the manuscript:

  1. An extensive editing for clarification and typos was made to this manuscript

Line 146-148, this is a provocative sentence that is not supported by the previous introduction.  Its not necessary either to set up the paragraph.

  1. This has been removed from the document

The first line of the paper reads as if the students were doing sexual violence to the academic institutions. It also is not supported by the content of the rest of the paragraph. There are statements about risk and prevalence of sexual assault for LGBTQ+ people which would appear to show that there is attention to this population in the literature.   This confusing first sentence sets the tone for the introduction – I don’t have a clear idea what the paper is about.  I think you should delete the sentence.

This sentence has been removed from the manuscript

Line 93 – be careful about changing terms. You have used post-secondary academic settings, educational institutions, then this heading uses Education.  Later you use school violence – this term refers to a different location and also a different form of violence (sexual violence vs school violence)

The authors have changed the language to be consistent throughout the document. 

Lines 93- 144 – this section should come out of the intro.  Some of the content could be moved to the discussion, if it is applicable to your findings.   This content provides background as to why people might not report.

The reason I recommend taking out this section, is because it is confusing the reader.  It takes the reader down a different road from the focus on college students and sexual assault.  This section is about high school and bullying.  It just confuses the reader as to what this study is about.  You are using a new term in this section too “school violence” and like I said, that takes the reader down a new road and is left unclear what the paper is about.

We have moved sections around and tightened up our literature review to be more cohesive. 

Line 66 – avoid words like “alarming.”  Conventionally, scientific writing avoids adjectives and adverbs like alarming.  Words like this are editorial and introduce some bias.  

  • The wording has been changed to high as to not have bias but still inform the reader of its prevalence.

Like 76-78 – this sentence can not be supported by the data and it also isn’t clear to the reader what you mean.  Are you saying that because so much attention has been on visibility of LGBTQ+ people that lack of data on gender or sexual orientation should be interpreted as everyone was cisgender?  Because if people were not cisgender then for sure their gender or sexuality would have been included?  That is a statement that you can’t support and so I think it should come out.

This statement was removed from the manuscript

Line 198 – 209 should be deleted.  It just adds to lack of clarity about what the study is about.

These lines were removed from the document

Line 209 – Clearly state the purpose of the study and have the same clear statement in the abstract.

The purpose has been edited to be more concise

Line 217 – This should be a statement of design, such as this study utilized a descriptive cohort design. 

In methods, labels interchange between sexual violence and sexual assault.  Sexual violence might be a better term – offensive comments are hurtful but not really an assault. 

The authors changed to verbage to only be sexual violence except for the analyiss of the question about being “sexually assaulted”. The authors have also added the utilization of a cohort design. 

One limitation is the question, did they participant report to campus officials or police.  Hurtful comments probably wouldn’t get reported to police.  Students might report to a faculty member or advisor, but would someone answering that survey consider a teacher to be a campus official? I think there is some limitations with the survey itself  in terms of the wording of the questions.  And also in calling all of the violent experiences sexual violence.  Also, did they have the option for “other” as a choice in their list of reasons to not report? Did they give reasons for why they reported? 

They did not have an option for other. The authors have included that some forms of violence may not be reported to the police as a limitation. 

Also consider the types of violence differently – the categories include instances of harassment/ bullying and partner violence. 

The questions that were analyzed had some form of sexual harassement or sexual tones, such as commenting on one’s body, sexual remarks, etc. This forms of interaction would fall into the overall theme of sexual violence. 

Results –

This section needs to get tighter too. 

In table 2, the Total column should show the number and percent of students who had this experience out of the total 808 students.  Presenting the data in that way shows that about around 10% of students had the first 3 things happen to them and lower percent had the last 3 things.

Tables were created to show age, race, sexual orientation, and gender characteristics. 

Are you able to report how many individuals had one of the types of violence? Some people might have experienced more than one type.   Of the 8?8 how many people had experienced at least one type of violence?

We did not single out whether an individual experienced multiple situations. We were concerned just about the number of each individual incident occurring in each category. We may have individuals that have multiple or singular incidents but would not retract from the total number of incidents that occurred. 

Discussion –

In your study only 4% reported sexual assault, this is lower than the 13% you have in the first sentence of the discussion.  Restatement of results is ok, but so what?  What are some measures that readers could or should take to use your results in their own work?  What are the implications of the findings?

That sentence was removed to not confuse the reader or take away from the importance of the study. 

You have an excellent topic and important data.  What does it mean for people at post-secondary institutions? What is being done to make things better?  What could be done to make things better?

We have added implications onto our conclusion

Reviewer 2 Report

The topic is of interest but the paper needs to be rewritten as it needs:

1. to follow academic writing style in the sciences.

2. to clarify the data and perhaps further data analysis.

3. to modify the conclusion with regards to the generalizability of the study.

Here are som some specific comments but they are not exhaustive. The authors should review the manuscript to correct all typographic and other errors, including those not addressed below:

Specifically,

1. The writing style doesn't clearly follow IMRAD. Specifically, while important, lines 198-209 should not fall under "purpose". Lines 253 -290 while appropriate in the methods section, should not be included under "data analysis". Further, much of that information appears in Table 1 hence, the information is redundant.

The first paragraph on limitations don't really address the important limitation of the study - the specific population recruited would be a select group. As such, the results of the study may not apply to general college students.

2. I am quite confused as to the study design. 1. How many schools were part of the target population? Howe many schools participated? Were the students from the various schools, all eligible to participate? Was participation in the scholarship application based on need? Were only LGBTQI eligible to participate? Were all participants invited to complete the survey or was it only LGBTQI participants? If heterosexuals were also invited to participate, why wasn't the analysis comparative (LGBTQI vs. heterosexual) since heterosexuals also experience violence.

Table describing the sample needs to be provided (demographics, etc). The authors state that separate scholarship were available for colleges/graduate students etc.) therefore, is the n=808 summative (aggregate) of the various students? If so, it would be important to show the constituents.

Please review the information posted. Line 243 and 244 have different frequencies for two-spirited people.

Lines 293/94. The analysis conducted is not clear. Would it be "experienced violence and reported vs "experienced violence and did not report"? The difference between the two groups would be based on other characteristics (variables).

The results as written are not clear. I could not follow, using my own  calculations as data is missing. All the importation should be present to help the reader make an informed conclusion.

3. By providing the pertinent and clear information on the study design, recruitment strategy, the merit of the study can then be put into perspective.

Author Response

Thank you for the thoughtful feedback from the reviewers. Based on the reviewers’ recommendations, edits have been made to the submitted document as outlined below. The researchers again would like to thank the editorial board for allowing the researchers to make revisions and resubmit. Below I have outlined the specifies on how the comments of the reviewers were changed in the manuscript:

The topic is of interest but the paper needs to be rewritten as it needs:

  1. to follow academic writing style in the sciences.
  2. to clarify the data and perhaps further data analysis.
  3. to modify the conclusion with regards to the generalizability of the study.

Here are som some specific comments but they are not exhaustive. The authors should review the manuscript to correct all typographic and other errors, including those not addressed below:

Specifically,

  1. The writing style doesn't clearly follow IMRAD. Specifically, while important, lines 198-209 should not fall under "purpose". Lines 253 -290 while appropriate in the methods section, should not be included under "data analysis". Further, much of that information appears in Table 1 hence, the information is redundant.

Lines 253-290 was moved to Study Design. Lines 198-209 have been moved to the previous paragraph before the purpose of study. 

The first paragraph on limitations don't really address the important limitation of the study - the specific population recruited would be a select group. As such, the results of the study may not apply to general college students.

This was added to the methods to clarify that it was a descriptive cohort so participants were chosen based on the identifying characteristic. 

  1. I am quite confused as to the study design. 1. How many schools were part of the target population? Howe many schools participated? Were the students from the various schools, all eligible to participate? Was participation in the scholarship application based on need? Were only LGBTQI eligible to participate? Were all participants invited to complete the survey or was it only LGBTQI participants? If heterosexuals were also invited to participate, why wasn't the analysis comparative (LGBTQI vs. heterosexual) since heterosexuals also experience violence.

The scholarship is an open scholarship that is the student, not through a specific institution. The scholarship is only for LGBTQ students. 

Table describing the sample needs to be provided (demographics, etc). The authors state that separate scholarship were available for colleges/graduate students etc.) therefore, is the n=808 summative (aggregate) of the various students? If so, it would be important to show the constituents.

Please review the information posted. Line 243 and 244 have different frequencies for two-spirited people.

This has been changed. 

Lines 293/94. The analysis conducted is not clear. Would it be "experienced violence and reported vs "experienced violence and did not report"? The difference between the two groups would be based on other characteristics (variables).

It is individuals who have only experienced sexual violence and out of those individuals, why did they not report. The authors clarified this more in the methods so readers can better understand. 

The results as written are not clear. I could not follow, using my own calculations as data is missing. All the importation should be present to help the reader make an informed conclusion.

We have added more tables to show the participant demographics to clarify information in the manuscript. 

  1. By providing the pertinent and clear information on the study design, recruitment strategy, the merit of the study can then be put into perspective.

Round 2

Reviewer 1 Report

The authors did a great job tightening up the intro section.  Each section is on topic and builds towards the purpose of the study. 

I appreciate the description of the survey- I understand how the survey was administered and to whom. 

Results are well organized.  I have a suggestion for rewording a few lines, my suggestions are below. 

There are still typos sprinkled throughout.  I identified some of them below.  There are typos in the punctuation, maybe that happened when the manuscript was put into a proofing format. 

Pg 1 - line 46 - I think the word cishet is a typo.

Line 156 - missing word after enhanced

Line 258 - missing the word "were"  only the e is visible

Line 299 "There were 92.2% (n=77) of those who had experiences this type of sexual violence did not report."  This sentence and the subsequent sentences like it (lines 304 & 309, 317, 322) should be reworded to something like, "Of the 106 participants who experienced this type of sexual violence, 92.2% (n=77) did not report it."

Because as its written, it could be misunderstood that 92% experienced this type of violence. 

LIne 313 - performed is misspelled - in the text its preformed

Line 351 - word missing after didn't. Also its conventional to write did not instead of didn't

Lines 357-360 - run on sentence.  Looks like some punctuation might have been missed, or maybe a word or two is missing. 

Line 371 - rather than "no" action taken, should be "any" action taken. 

Line 377 - that should be changed to "and" - this sentence could also be broken into two sentences. Shorter sentences are my preference, as they are easy for the reader to understand. 

Line 378 - missing a word. The subsequent lines seem like are missing a word too.  

Read over the discussion for readability. 

Line 389 LGBTQ+ changes to LGBTQIA+ If you are changing the acronym, you need an explanation.  Or maybe you want to use LGBTQIA+ from the beginning?

Author Response

Pg 1 - line 46 - I think the word cishet is a typo.

            Cishet means cisgender heterosexual. I have made it more clear in the paragraph.

Line 156 - missing word after enhanced

            Removed due to so it reads “…sexually assaulted also experience enhanced internal feelings of shame of being a sexual minority…”

Line 258 - missing the word "were"  only the e is visible

            That has been added

Line 299 "There were 92.2% (n=77) of those who had experiences this type of sexual violence did not report."  This sentence and the subsequent sentences like it (lines 304 & 309, 317, 322) should be reworded to something like, "Of the 106 participants who experienced this type of sexual violence, 92.2% (n=77) did not report it."

            The results was rewritten to make it clear how many participants in total had experienced the type of sexual violence.

Because as its written, it could be misunderstood that 92% experienced this type of violence. 

LIne 313 - performed is misspelled - in the text its preformed

            This was fixed.

Line 351 - word missing after didn't. Also its conventional to write did not instead of didn't

            Now reads. “…participants just did not think any action would be taken..”

Lines 357-360 - run on sentence.  Looks like some punctuation might have been missed, or maybe a word or two is missing. 

            “In line with previous research, the present study also shows that there is a high prevalence of LGBTQIA+ students who have been a target of sexual violence and that with this high incidence of sexual violence. There is also a high prevalence of individuals not reporting the sexual violence regardless of the severity of the action.”

Line 371 - rather than "no" action taken, should be "any" action taken. 

            This was fixed.

Line 377 - that should be changed to "and" - this sentence could also be broken into two sentences. Shorter sentences are my preference, as they are easy for the reader to understand. 

Line 378 - missing a word. The subsequent lines seem like are missing a word too.  

            The authors went and addressed the readability of the discussion. 

Read over the discussion for readability. 

Line 389 LGBTQ+ changes to LGBTQIA+ If you are changing the acronym, you need an explanation.  Or maybe you want to use LGBTQIA+ from the beginning?

The acronym is changed throughout to only be LGBTQ+ since that was the population mentioned in the study.